# Bypassing Mendel’s First Law: Transmission Ratio Distortion in Mammals

**DOI:** 10.3390/ijms24021600

**Published:** 2023-01-13

**Authors:** Gaëlle Friocourt, Aurore Perrin, Paul A. Saunders, Elvira Nikalayevich, Cécile Voisset, Charles Coutton, Guillaume Martinez, Frédéric Morel

**Affiliations:** 1INSERM, Univ Brest, EFS, UMR 1078, GGB, 29200 Brest, France; 2Department of Medical Genetics and Reproductive Biology, Brest University Regional Hospital, 29609 Brest, France; 3School of Natural Sciences, University of Tasmania, Sandy Bay, TAS 7000, Australia; 4Center for Interdisciplinary Research in Biology (CIRB), College de France, CNRS, INSERM, Université PSL, 75231 Paris, France; 5Service de Génétique, Génomique et Procréation, CHU Grenoble Alpes, 38000 Grenoble, France; 6Institute for Advanced Biosciences, INSERM, CNRS, Université Grenoble Alpes, 38000 Grenoble, France

**Keywords:** transmission ratio distortion, meiotic drive, gametogenesis, fertility, *ARX*

## Abstract

Mendel’s law of segregation states that the two alleles at a diploid locus should be transmitted equally to the progeny. A genetic segregation distortion, also referred to as transmission ratio distortion (TRD), is a statistically significant deviation from this rule. TRD has been observed in several mammal species and may be due to different biological mechanisms occurring at diverse time points ranging from gamete formation to lethality at post-natal stages. In this review, we describe examples of TRD and their possible mechanisms in mammals based on current knowledge. We first focus on the differences between TRD in male and female gametogenesis in the house mouse, in which some of the most well studied TRD systems have been characterized. We then describe known TRD in other mammals, with a special focus on the farmed species and in the peculiar common shrew species. Finally, we discuss TRD in human diseases. Thus far, to our knowledge, this is the first time that such description is proposed. This review will help better comprehend the processes involved in TRD. A better understanding of these molecular mechanisms will imply a better comprehension of their impact on fertility and on genome evolution. In turn, this should allow for better genetic counseling and lead to better care for human families.

## 1. Introduction

In diploid organisms, hereditary characteristics are transmitted through generations and their segregation is ensured by meiosis and fertilization. According to the fundamental principles of Mendelian genetics, the two alleles of one diploid locus have an equal probability (50:50) of being transmitted to the offspring. Exceptions in which a statistical deviation from the 1:1 Mendelian inheritance ratio is observed are grouped under the term transmission ratio distortion (TRD) and occur when one of the two alleles of one of the parents is preferentially transmitted to the offspring. TRD has been observed in many mammalian species [1,2,3,4,5] and may be due to diverse biological mechanisms occurring at different time points ranging from gamete formation to lethality at post-natal stages (Figure 1).

TRD can first occur from a mechanism called germline selection (Figure 1) [6]. During the proliferative phase of gametogenesis, gonia (gamete precursors) mitoses lead to the production of a pool of genetically identical cells in which mutations, mitotic gene conversion and crossing-overs create genetic diversity [6]. The genetic variations generated can either confer a proliferative advantage to some of those cells or, in contrary, cause their elimination [6,7]. This mechanism is probably much more frequent in spermatogenesis than in oogenesis because the oogonia undergo less pre-meiotic mitoses than spermatogonia.

Another TRD mechanism consists in the non-random segregation of chromosomes during meiosis called meiotic drive, a term introduced by Sandler and Novitski [8]. Pardo-Manuel de Villena and Sapienza [3] suggested that a meiotic drive could occur if three conditions are met: (i) an asymmetric meiosis such as in oogenesis (where each oocyte produces only one functional gamete with the expulsion of two polar bodies); (ii) a functional polarity of spindle poles with a cortical side (“cortex pole”) and an egg side (“egg pole”); and (iii) a functional heterozygosity at a locus inducing a different interaction to the spindle. The meiotic drive involves selfish genetic elements that can involve centromeric DNA sequences (centromeric drive) or other sequences (non-centromeric drive).

During spermatogenesis, meiosis is symmetric (one spermatocyte producing four functional gametes), thus the meiotic drive is exceptional [1,9]. Indeed, when originating from the paternal side, TRD occurs more frequently through gametic competition. It is a form of post-meiotic selection taking place between the end of meiosis and fertilization, for example during spermiogenesis (the transformation of a spermatid into a spermatozoon) and/or during the free life of the spermatozoa. This mechanism leads to differential mobility, viability and/or fertilization ability between spermatozoa. Various models, such as the poison–antidote model or the killer–target model, have been observed [10]. Although no post-meiotic TRD has been reported in mammalian oocytes, it is notable that it has been observed in insects [11].

TRD can also be due to zygotic selection, differential embryonic/fetal lethality or even a post-natal lethality if the genotype is deleterious [12,13,14,15].

Finally, errors in resetting genomic imprinting in the parental germline or failure to maintain imprint occurring at fertilization or during the embryonic development can be responsible for TRD [16,17,18]. Faulty genomic imprinting may not be compatible with the survival of the zygote or the embryo.

In this review, we describe examples of TRD and their possible mechanisms in mice, humans and other mammals based on current knowledge.

## 2. Transmission Biases Reported in Mice

### 2.1. Asymmetric Meiosis

#### 2.1.1. Polarity during Female Meiosis

During oogenesis, an oocyte undergoes asymmetric meiosis where only one of four sets of chromosomes is preserved. During the first division, the homologous chromosomes separate from one another and one set (2n) is extruded in a small polar body, while the other set (2n) remains in the oocyte. During the second division, sister chromatids segregate, with one set (n) removed in the second polar body. The remaining set of chromosomes (n) is transmitted to a new organism after fertilization.

Both polar bodies are small in order to preserve the oocyte organelles, proteins and mRNA cytoplasmic stores, which are crucial to support early embryonic development. The polar bodies and the genetic material they contain are eliminated. This asymmetry of female meiosis, where only one of four sets of chromosomes is conserved, creates a favourable environment for the evolution of selfish genetic elements. Female meiotic drive can occur during the first or the second oocyte division. There is no evidence of a driving element capable of biased transmission during both divisions, except for the homogenously stained region (HSR) element, which is discussed below.

Chromosome segregation is driven by a microtubule bipolar spindle where each chromosome is connected to microtubules through its kinetochore assembled at the centromeric region. The spindle assembly checkpoint (SAC) ensures that by the time of anaphase, all chromosomes are attached to the microtubule spindle and that each homologous chromosome in the bivalent (in the first division) is oriented towards a different spindle pole. Before anaphase of the first meiotic division, the spindle is positioned perpendicular to the oocyte cortex with one pole being closer to the centre of the oocyte than the other. The chromosomes attached toward the egg pole will remain in the oocyte and the chromosomes attached toward the cortex pole will be expelled.

#### 2.1.2. Meiotic Drive in the First Meiotic Division

##### Centromeric Drive

Selfish elements can bias the usually random orientation of chromosomes inside the spindle to favour their retention in the future egg. A well-described example of these elements are selfish centromeres, or centromeric drive. Centromeres are regions of the chromosomes composed of small satellite repeats that bind specialized nucleosomes and serve to assemble kinetochores that will attach the chromosomes to the spindle and ensure their segregation during anaphase. There is evidence that larger centromeres have preferential transmission to oocytes rather than to polar bodies during the first meiotic division [19] due to the asymmetry of the spindle [20,21]. 

The meiotic spindle forms in the centre of mouse oocytes around the chromosomes [22,23,24]. At this point, there is no distinguishable egg pole or cortex pole, and larger centromeres have no bias in their attachment to either pole [20]. A few hours before anaphase I, the spindle migrates to the cortex [24], where signalling from the chromosomes causes the closest area of the cortex to polarize and signal back to the spindle, leading to the cortex pole microtubules’ tyrosination level to rise. Tyrosinated microtubules are more unstable, which helps selfish centromeres to differentiate between poles [20]. Selfish centromeres recruit more of the microtubule-destabilizing proteins, which normally are a part of the SAC [25]. When selfish centromeres are facing the cortex and the tyrosinated half of the spindle, they trigger iterative detachment and reattachment events. The reattachment can happen to any pole, so after several detachment and reattachment events, the selfish element can turn to the egg side of the spindle where the microtubules are more stable. This activity can delay anaphase and it has indeed demonstrated that oocytes with centromeric drive can have longer metaphase I [20].

Centromeric sequences are among the fastest evolving elements, possibly due to selection pressure during female meiosis [26]. To avoid a rapid and possibly deleterious spread of selfish centromeres, proteins recruited to the centromeric drive are rapidly changing too. The “arms race” model suggests that centromeric DNA evolves through the strengthening of its interaction with proteins that directly bind to it but the affinity of these proteins to centromeric sequences is simultaneously weakening [26]. A more recently proposed “parallel pathways” model suggests that, in addition to the decreased affinity of core centromeric proteins to centromeres, the mechanism of kinetochore assembly shifts to increase pericentromeric heterochromatin involvement [27]. This parallel modification of the pathway of kinetochore assembly would limit the selfish centromere capacity to assemble bigger kinetochore and reduce the impact of its microtubule destabilizing activity.

Whether or not centromeric drive can affect the chromosomes’ fate during the second meiotic division is still unclear, as both centromeres of the sister chromatid pair are identical. Artificially altering the centromere strength of only one centromere of the pair might allow us to answer this question.

##### XO Mice

Another example of distorted transmission occurring during the first meiotic division is the X chromosome segregation in XO female mice that lack a second sex chromosome. Unlike humans, where females with a single X chromosome are sterile, the XO mice are fertile. If the segregation of a single X chromosome was random, these females would produce daughters with an XX or XO genotype in equal proportions. However, the XO females preferentially produce XX daughters, suggesting biased transmission of the single X chromosome to the egg [28]. 

In XO oocytes, the single X chromosome has no pairing chromosome and has been demonstrated to have a higher chance to remain in the oocyte after the first meiosis [28]. It remains to be determined whether this transmission bias might be similar to the centromeric drive.

In some rare cases, a single X chromosome can split into sister chromatids prematurely during the first meiotic division, and each chromatid segregates into the oocyte and the first polar body. Then, the remaining single chromatid seems to segregate randomly during the second division [28], suggesting that the mechanism of the X chromosome TRD in XO oocytes is specific to the first meiosis.

#### 2.1.3. Meiotic Drive in the Second Meiotic Division

Biased transmission of genetic elements has also been demonstrated in the second meiotic division and is sometimes referred to as “non-centromeric drive”, as the mechanism of this drive is still elusive. The spindle in metaphase II is parallel to the cortex and does not have a distinct egg side or pole side. After anaphase II initiation, chromosomes segregate and are pulled towards the spindle poles, each chromatin mass producing local centers of polarization in the cortex [29]. Cortex polarization shift destroys spontaneous symmetry in the forces acting on the spindle, causing spindle rotation [30,31]. Which pole will turn towards the cortex and will be extruded is considered random. It would, however, be interesting to confirm these findings in oocytes presenting genetic elements that are driving in the second meiotic division.

##### R2d2

The R2d (responder to drive) is a 158 kb section located in the R2d1 locus on chromosome 2. Certain mouse strains also have an R2d2 locus, which is a high-copy duplication of R2d [32]. The R2d copy number in the R2d2 locus varies between strains and can reach up to 35 elements. The R2d2 locus contains multiple copies of the Cwc22 gene, which encodes a splicing factor that has been demonstrated to be expressed and functional in R2d2-bearing organisms. The R2d2 locus influences its own transmission during female meiosis, but only if it is present at a high copy number, as low-copy number R2d2 females display no transmission bias [32]. Because the transmission bias depends on the number of the R2d sequence repeats, it has been suggested that the locus may act as a neocentromere [32].

This element is considered to have originated from a single event and has spread rapidly to numerous wild populations of mice even though it is not beneficial. For example, R2d2 has a deleterious effect on the litter size in heterozygous females with transmission distortion [33]. 

##### Ovum Mutant

The ovum mutant (Om) locus is a region on chromosome 11 carrying a factor responsible for high rates of embryonic lethality. Om locus is associated with the “DDK syndrome” (dysdiadochokinesia), named after a strain for which mating DDK females with males of other genetic backgrounds results in litters exhibiting over 95% embryonic lethality [34]. The region on chromosome 11 carrying the Om allele segregates with transmission bias in female meiosis [35,36] during the second meiotic division only [2,37]. Curiously, loci flanking the Om locus also segregate with a transmission bias, albeit with lower efficiency [35,36]. 

##### HSR

The homogenously stained region (HSR) is located on chromosome 1. It consists of a series of long-range repeats [38]. A variant of chromosome 1 bearing an extra insertion of HSR has been shown to have a strong meiotic drive during the second meiotic division [39]. Like the R2d2 locus, it is suggested that the driving HSR may form a neocentromere due to the presence of multiple repeats.

Some evidence points to biased transmission of the HSR element on the non-crossover chromosomes during the first meiotic division [39]. As HSR is located distally to the centromere, the mechanism of this bias may differ from the centromeric drive. Another feature of HSR segregation in oocytes includes a possible influence by the sperm, discussed in more detail elsewhere [40].

To summarize, non-Mendelian chromosome segregation in mouse oocytes is possible due to the asymmetric nature of female meiosis that leads to retention of only one out of the four chromosomes. The centromeric drive in the first meiotic division acts through destabilizing chromosome attachment to the cortical pole of the spindle and favors chromosome attachment to the egg pole. The meiotic drive in the second meiotic division depends on driver elements located on chromosome arms, but the mechanism of their segregation bias remains to be deciphered.

### 2.2. Symmetric Meiosis

Male meiosis is very different from female meiosis and might appear less prone to transmission biases at first sight. As opposed to the asymmetric nature of female meiosis, male meiosis is symmetric: it results in the production of four functionally equivalent haploid sperm cells. Therefore, there is no opportunity for selfish genetic elements to gain a transmission advantage by ensuring their non-selfish counterparts end up in non-fertilizing cells (akin to polar bodies). Furthermore, the developing male germ cells are connected to neighboring cells via cytoplasmic bridges, allowing for cells to share gene products. These bridges supposedly evolved to help synchronize the development of spermatids and make haploid sperm cells transcriptionally diploid [41,42,43]. This should erase any phenotypic difference emerging from haploidy (e.g., differences between X- and Y-bearing spermatids or spermatozoa), preventing selfish genetic elements from recognizing self from non-self to promote their own transmission.

Nevertheless, some male transmission distorters have been characterized in plants and animals over the years—including several in mice—and a few general principles have emerged from their study (for a recent comprehensive review, see [10]). In a symmetric meiosis, a transmission distorter must be able to detect phenotypic differences between spermatids carrying the distorter and those that do not and exploit these differences to generate a contrast between their viability or fertilizing ability in order to gain a transmission advantage. All known examples seem to occur post-meiotically through the harming of the non-carrying sperm cells. Theoretically, the simplest way for a genetic locus to achieve this is through the production of two factors (RNA or protein): a driver/distorter factor that would harm spermatids and a responder factor that would ensure that only the non-carrier spermatids or sperm cells are affected. The first factor, also referred to as a killer or a poison, is shared through cytoplasmic bridges, while the second must somehow manage to avoid sharing: it must be expressed solely in non-carrying cells and act as a target for the killer driver; or solely by the carrying cell and act as an antidote that prevents poisoning (Figure 2). Another crucial requirement is that the genes coding for these factors need to be tightly linked genetically, as recombination could lead to generating “suicidal” variants (e.g., carrying the driver but not the antidote).

#### 2.2.1. T-Haplotype

The t-haplotype is a naturally occurring variant of mouse chromosome 17 that has been studied for almost a century, primarily because it cheats Mendel’s first law: males heterozygous for a t-haplotype and a wild-type chromosome 17 (t/+ males) transmit the t-haplotype to up to 99% of their offspring (see [44] for an extensive review of its discovery and early work). The t-haplotype consists of a large 40 Mb region that is inherited as a single unit due to multiple chromosomal inversions, and a large body of evidence suggests it is a prime example of a poison–antidote system. The study of multiple crosses of mice with different versions of the t-haplotype (complete or truncated) revealed that drive is due to the cumulative effect of multiple driving genes (initially called Tcd) acting on one responder gene (Tcr): only in the presence of both Tcd and Tcr is the t-carrying chromosome advantaged over the wild-type one [45]. In parallel, the physiological mechanism of the drive was uncovered. Spermatogenesis proceeds normally in t/+ males, and their ejaculates contain a balanced ratio of t- and +-bearing sperm [46]. However, the analysis of sperm flagellar function revealed that +-bearing sperm show altered motility (they are more vigorous, but non-progressive), providing the t-bearing sperm an advantage in reaching the fertilizing site in the female reproductive tracts [5,47]. The responder gene called *SmokTcr* was isolated in 1999 [48]. It is part of the sperm motility kinase (Smok) multi-gene family that are expressed in late spermatogenesis and are involved in the proper establishment of sperm motility via phosphorylation of the sperm flagellar axoneme. As predicted by the poison–antidote model, SmokTcr transcripts escape sharing between neighboring cells and are restricted to t-sperm cells [49]. This contrasts with the four driving genes identified so far: *Tagap1*, *Fgd2*, *Nme3* and *Tiam2*, which are expressed pre-meiotically and whose gene products are shared in all cells of the syncytium [50,51,52,53]. These four genes synergistically hyperactivate wild-type SMOK activity in all sperm cells, which leads to impaired sperm motility only in +-bearing sperm, as SMOKTCR counterbalances this hyperactivation in t-bearing sperm [5,50]. Similar to many other transmission distortion systems segregating in natural populations, the t-haplotype seems to be maintained at an intermediate equilibrium frequency (10–25% [54]). The reason it does not spread to fixation seems to be two-fold: males homozygous for the t-haplotype are sterile because of the presence of recessive deleterious mutations within the 40 Mb non-recombining region [44] and in heterozygous males, in addition to harming +-bearing sperm, the t-haplotype seems to also harm (to a lesser extent) t-bearing sperm [55], resulting in a low success of t/+ males in sperm competition with +/+ males.

#### 2.2.2. Sex Chromosome Drive and Sly/Slx Genes

Another major difference between female and male meiosis in mammals is that the latter involves the segregation and transmission of the X and Y chromosomes. Since recombination is suppressed in the heterogametic sex in many plants and animals with sex chromosomes, they are thought to be an ideal spot for the evolution of transmission distorters that require the interaction between tightly linked genes [56,57]. However, the biased transmission of sex chromosomes (also called sex chromosome drive) results in biased sex ratios, which goes against the general tendency in animals to produce 1:1 male and female offspring and is highly non-adaptive from an autosome’s perspective [58]. Such conflicts lead to strong selection against drive and should result in the rapid emergence of drive suppressors on either or both the non-driving sex chromosome or/and autosomal loci [59]. Over time, an evolutionary arms race between the driver and suppressor(s) can emerge, causing rapid evolution of genetic sequences and/or alternated bouts of gene duplication if the loci involved have antagonistic effects [60,61].

The multicopy genes Sly (Sycp3-like Y-linked) and Slx/Slxl1 (Sycp3-like X-linked and Slx-like 1), found on the Y and X chromosomes, respectively, in approximately 50 to 150 copies in different house mouse populations, are engaged in such a conflict [62,63]. The copy number of Sly and Slx/Slxl1 in natural populations shows a strong correlation [63] and functional studies of transgenic mice have revealed that a balanced copy number is necessary for the equal transmission of X and Y chromosomes. Unbalance has several detrimental consequences: in Sly-deficient males, spermatogenesis is altered (leading to sperm cell defects and reduced fertility) and sex ratios are female-biased due to a biased transmission of the X chromosome [64]. The severity of fertility loss and the magnitude of the X-drive is proportional to the number of Sly copies [64,65]. Slx/Slxl1-deficiency also alters spermatogenesis but has the opposite effect on sex ratio: males with missing copies of Slx/Slxl1 produce more male offspring due to a biased transmission of their Y [66,67]. Interestingly, simultaneously knocking down Slx/Slxl1 and Sly restores a balanced X:Y transmission ratio [64]. Like the t-haplotype, biased transmission is due to abnormal sperm development. In mice with an unbalanced ratio of Sly vs. Slx/Slxl1 in favor of the latter, sperm cell morphology and motility are altered, though more severely in Y-bearing sperm [68]. This makes X-bearing sperm more likely to reach the eggs first, hence the female-biased sex ratio. An unbalanced copy number in favor of Sly also results in sperm cell defects [67], and logically, X-bearing sperm should be more severely impacted to account for the male-biased sex ratio.

How do Sly and Slx/Slxl1 impact sperm development and sex chromosome transmission? It has been proposed that they are in competition and that the conflict is dosage-dependent, i.e., mediated by their expression level. Both sets of genes are expressed exclusively in spermatids, post-meiotically. They code for proteins that regulate the expression of hundreds of other genes: primarily sex-linked and mostly expressed during spermatogenesis, some of which are crucial for proper sperm cell development [65,67,69,70]. Sly and Slx/Slxl1 are both found in X- and Y-bearing spermatids and compete for the same genomic targets. However, they have antagonistic effects: Sly acts as a down-regulator of gene expression, while Slx/Slxl1 increase expression, which explains why a balanced copy number is required for unaltered spermatogenesis (For details on molecular mechanisms, see [67,71,72]). Among the hundreds of genes regulated by Sly and Slx/Slxl1, it has been speculated that one or several differentially impact the fitness of X- versus Y-bearing sperm, but their identity remains elusive. Notably, several *Smok* genes (involved in the t-haplotype transmission distortion) are found among the target genes, highlighting a potential functional relation between the two drive systems [72].

#### 2.2.3. Other Male Transmission Distortion Systems in Mice

In the 1990s, transmission distortion was described in several lines of mice carrying different Robertsonian fusions (Rb(6.16) and Rb(6.15)): in heterozygous males, the fusion is transmitted to over half of the offspring [73,74]. Like the t-haplotype and Sly/Slx drive, males produce a balanced number of Rb/wild-type sperm suggestive of a post-meiotic drive mechanism. Distortion likely involves several genes from the proximal region of chromosome 6: *Spam1* (sperm adhesion molecule 1) and a cluster of *Hyal* (hyaluronidase) genes [75,76]. They code for proteins located in the sperm membrane, essential for fertilization, and the Rb alleles carry several mutations in strong genetic linkage due to the reduced recombination induced by the fusion. Spam1 is a particularly interesting candidate, as it escapes sharing across the cytoplasmic bridges between neighboring spermatids and the Rb alleles show decreased levels of transcript and protein expression, generating differences between the wild-type and Rb sperm [75,77]. To the best of our knowledge, a causal explanation between the decreased expression of Spam1 and biased transmission has yet to be proposed.

Recent work suggests that X-linked Toll-like receptor genes *Tlr7* and *Tlr8* could also be involved in a transmission distortion system [78]. Their gene products are not shared across sperms in the syncytium and, as a consequence, only X-bearing sperm carry these surface receptors. In vitro pharmacological activation of TLR7/8 significantly reduces X-sperm motility, which should theoretically lead to a lower transmission rate if those sperm cells encounter a similar ligand in naturae. More work on this system is needed, in particular to understand why TLR7/8 are not shared with Y-bearing sperm cells, but it has been proposed that these receptors could act as a target for a Y-linked killer that still has to be identified [79]. 

Finally, sex chromosome drive has also been described in several other rodents. These rodents have something in common: unusual sex determination systems, in which either the Y has disappeared, or a third sex chromosome (homologous to a bird W chromosome) has emerged (reviewed in [80]). For instance, male sex chromosome drive was recently described in the African pygmy mouse *Mus minutoides*, a very close relative of the house mouse, in which three sex chromosomes coexist [81]. These rare deviations of the standard XY sex determining system are thought to facilitate the spread and maintenance of sex chromosome transmission distortion, and though nothing is known about the molecular or physiological mechanism involved in these systems, it is not impossible that they share similar gene networks and targets with the other drive systems described above, and they represent valuable models to obtain a better understanding of these mechanisms.

## 3. Transmission Biases Reported in Other Mammals

Transmission ratio distortion has been extensively studied among a variety of taxa in insects [82,83,84,85], birds [86,87,88], fish [89,90] and crustaceans [91], but very few mammalian species have been investigated beyond man and mouse. Despite the significant scientific interest in studying TRD phenomenon in species with higher genetic proximity to humans, most species documented to date, except for the common shrew [1,92,93], are livestock species, such as cattle [5,16,94,95,96,97], pig [4,98,99,100,101,102], horse [103,104], goat [105] and sheep [106]. The interest in these taxa originates mainly from the agricultural industry for whom any genetic factor, such as TRD that can influence the reproductive success of animals, is of major interest in terms of production objectives [107]. Also, the fact that we possess complete genealogical follow-ups through several generations with genetically identified relatives for these species is an undeniable advantage for TRD studies, with each introduction of new genetics in the herds (to improve the stock or to avoid inbreeding) being strictly referenced [108].

Preliminary evidence of TRD in livestock was reported as early as 1985 with the segregation study of eleven swine leukocyte antigen (SLA) haplotypes in two breeds of pigs [98]. Inspired by this report, as well as others on mice and humans [109], excess male transmission of a major histocompatibility complex haplotype was quickly identified in horses [104]. A subsequent study of alleles at twelve protein marker loci in a large cohort of more than 5000 phenotyped horses reported the first sire-specific gene TRD that had taken over all typed loci [103]. The involvement of the cattle industry led to the identification of several TRDs resulting from different events occurring before, during or after fertilization. For example, as sexing of cattle is of significant economic interest to breeders, Szyda et al. [94] examined the segregation of sex chromosomes in 2122 sperm from 35 bulls and found a higher proportion of gametes carrying an X chromosome (53 ± 5%). Without formal demonstration, this TRD was suggested to result from pre-fertilization events during meiosis. Interestingly, the observed sex ratio was linked to the variation in the recombination rate among X-bearing sperm cells, but not among Y-bearing spermatozoa. Furthermore, in 2021, Amaral et al. uncovered the important role of RAC1 in progressive sperm cell motility by extending the study of the well-known mouse t-haplotype (see corresponding part) to cattle [5]. Sperm cells carrying the t-haplotype have a more progressive movement than those carrying the wild-type haplotype and are therefore very preferentially (>99%) transmitted to the offspring by systematically winning the race to fertilization. Finally, a post-fertilization TRD was identified when Murphy et al. [95] demonstrated a deviation in the expected genotype ratios for the GH1 p.Leu127Val polymorphism during early embryo development in cattle in favor of the GH1 Leu127 allele. This polymorphism is of great interest, as it was associated with milk production and oocyte maturation in females and sperm production in males. Since this deviation is only observed in embryos produced in vitro and not in those produced in vivo, it was assumed that it originates from the less favorable environment of in vitro culture.

These studies used classical methods of TRD analysis involving standard mating patterns, such as backcrossing (comparing degrees of TRD observed in offspring carrying parental and non-parental chromosomes to determine whether it originates from meiotic or post-meiotic events), or on F2 populations [2,110]. A breakthrough occurred in 2014 with the implementation of Bayesian models for genome-wide analyses of TRDs within a given biallelic genetic marker, allowing for the investigation of populations of more diverse structures [101]. This model, initially limited to biallelic loci, was refined and Bayesian TRD models for haplotypes were implemented [97], leading to the development of a useful free statistical software for genome-wide searches for TRD regions (TRDscan v.1.0, http://www.casellas.info/files/TRDscan.zip, accessed on 22 August 2022). It is worth noting that TRD scans’ performance remains subject to the quality of SNP calling, and interpretation thresholds have been suggested to eliminate doubtful signals, especially in cases where trios are incomplete [105]. Also, implementation and comparison of different parametrizations seems relevant to accurately capture both allele- and genotype-related TRD [15]. Nevertheless, this led to the emergence of large screening studies of TRDs in farmed species.

For cattle, Casellas et al. [96] characterized the distribution of TRD by screening sire- and dam-specific TRD phenomena across 602,402 single nucleotide polymorphisms on 168 families (parent-offspring) from seven Spanish beef cattle breeds. They reported a preponderance of sire-specific TRD (0.13%, *n* = 786) when compared with dam-specific TRD (0.01%, *n* = 29) with no SNP accounting for any of them. However, a high degree of heterogeneity was evidenced among the breeds, with a single sire-specific marker (rs43147474) recurring in all breeds. For swine, the genome of both paternal and maternal TRDs was recently investigated. Casellas et al. [101] uncovered 84 SNPs displaying significant paternal TRD from an analysis of 29373 SNPs on 5 males and their 352 offspring. Godia et al. [102] sequenced both diploid and haploid genomes from three males and identified 55 SNPs displaying TRD. Most of them were animal-specific, but two genes and four genomic regions were common to all boars. Finally, Vázquez-Gómez et al. [4] scanned the genome of 247 Iberian pig families (dam-offspring) from two distinct varieties. They reported 68 and 24 maternal TRD loci, respectively, from 16246 and 9744 SNPs analysis in Entrepelado and Retinto populations, ten of which were common to both. In all three studies, many of the regions involved in TRDs contained genes implied in biological processes associated with transcription, spermatogenesis, embryonic viability, placenta development and diverse disorders. For sheep, Huang et al. [106] focused on parents–offspring trios to investigated potential TRDs of specific MHC class II haplotypes in a wild population of Soay sheep (*Ovis aries*). They reported some evidence of paternal distortion of one of the MHC haplotypes, although this could not be statistically confirmed.

Regarding TRD involving chromosomal translocations, the common shrew (*Sorex araneus*) is probably the most fascinating model for cytogenetic studies. Indeed, this species possesses one of the most remarkable chromosomal polymorphisms ever recorded. The common ancestor of modern shrews had a karyotype composed exclusively of acrocentric chromosomes and the occurrence of Robertsonian fusions induced a very strong evolutionary divergence that resulted in more than 50 distinct chromosomal races of shrews nowadays [111]. The predominance of metacentric chromosome (produced by the Robertsonian fusion of two acrocentric chromosomes) inheritance has been extensively investigated and three studies [1,92,93] reported what is undoubtedly a TRD with both a maternal and paternal preferential inheritance of metacentric over acrocentric chromosomes (0.6–0.7). While the mechanisms responsible for the spread and retention of metacentrics in this species remain uncertain, the reported effects appear to be individual and sex-dependent with a stronger effect in males [1]. More recently, a study investigated TRD in pigs [100] carrying a reciprocal translocation t(13;17) and reported paternal and maternal meiotic segregation results. They found that the females displayed more unbalanced gametes (28.92% versus 3.22%) due to the production of more diploid gametes and unbalanced gametes from adjacent segregation. Even if it remains speculative, the origin of the distortion in this context probably lies in the more permeant meiotic checkpoints of female meiosis and its proneness to more meiotic errors [112,113,114,115].

## 4. Transmission Biases Reported in Human and in Pathologies

### 4.1. Robertsonian Translocations

Robertsonian translocations in humans may induce a risk of infertility, recurrent miscarriages, stillborns or the birth of children with unbalanced karyotypes [115,116,117]. Indeed, carriers may have impaired gametogenesis and/or may produce chromosomally unbalanced gametes [118,119]. During gametogenesis of these carriers, three types of meiotic segregation occur: the alternate mode produces gametes that are either chromosomally normal (with the two acrocentric chromosomes) or balanced (with the translocated chromosome) in equal rate, while adjacent and 3:0 modes produce chromosomally unbalanced gametes.

Pardo-Manuel de Villena and Sapienza [3] and then Daniel [120] found that, in male carriers, the 50/50 ratio of normal versus balanced offspring is respected. However, female carriers show a TRD in favor of balanced offspring. This TRD is due to non-random segregation of chromosomes during the first meiotic division. Indeed, the centromeric or pericentromeric fusion increases the DNA satellite repeats of the translocated chromosome, which is thus transmitted preferentially to the oocyte rather than to the first polar body (see Section 2.1.2).

### 4.2. TRDs in Pathologies Caused by Point Mutations

A transmission bias has also been reported in genetic diseases caused by point mutations. For example, pseudohypoparathyroidism type 1A (PHP1A) and pseudopseudohypoparathyroidism (PPHP) are two dominant genetic diseases caused by rare loss-of-function mutations in the GNAS (Guanine Nucleotide Binding Protein, Alpha Stimulating Activity) gene, which encodes the α-subunit of the stimulatory G protein (Gsα). PHP1A is due to mutations in the maternal allele and results in Albright’s hereditary osteodystrophy (AHO) with hormonal resistance, whereas PPHP, characterised by AHO features but without hormonal resistance, is due to mutations in the paternal allele. These two different phenotypes are explained by imprinting of the *GNAS* gene, and thus of the expression of Gsα, which is different depending on the sex of the transmitting parent. For both diseases, a TRD has recently been shown with an excess of transmission of mutated alleles from the mother to the offspring, whereas in contrast, a Mendelian distribution was observed when the mutations were paternally inherited [121], suggesting a role of Gsα in oocyte biology or embryogenesis. Indeed, several studies have shown that AMPc levels and the AMPc pathway-protein kinase A are involved in the regulation of female meiosis in mammals, playing a role in both ovaries by controlling the meiotic prophase 1 in oocytes, folliculogenesis and asymmetrical division [122,123,124,125]. Since *GNAS* is maternally expressed in the ovaries [126], this could affect the transmission of this allele to the offspring. 

Another example of TRD in humans is the long QT syndrome, a cardiac conduction defect that increases the risk of syncope and sudden death, and which occurs in one in 2500–5000 births. Thirteen candidate genes are implicated, of which two are potassium channel genes (*KCNQ1*, *LQT1* [MIM 192500] and *KCNH2*, *LQT2* [MIM 613688]). The main diagnostic feature of long QT syndrome is the duration of the QTc interval. Women may be more likely to be diagnosed because, physiologically, their QTc interval is longer than that of men. However, the question arose as to whether the observed higher incidence in women is real or is due to a recruitment bias. A first study revealed in 2006 that classic Mendelian inheritance ratios were not observed in the offspring of female carriers of *LQT1* or male or female carriers of *LQT2*, with a higher proportion of genetically affected offspring than expected: the disease-causing allele was transmitted more often to female offspring than to male offspring and increased maternal transmission of the long QT syndrome mutations to daughters was also observed, possibly contributing to the excess of female patients with this disease [127]. These observations were confirmed 10 years later by Itoh and colleagues [128], who further showed that preferential allele transmission was linked to the degree of ion channel dysfunction, at least for *KCNQ1* variants. As this gene is expressed in ovaries, granulosa and trophoblastic cells [129], mutations could impact early development and/or fertilisation in which ion channels are of crucial importance, favouring maternal transmission.

### 4.3. TRDs in Pathologies Caused by Trinucleotide Expansions

Cases of TRD in humans have only rarely been reported, probably because of the difficulty to detect them due to the small size of the families and/or genetic counselling that may prevent parents from having several affected children. In addition, it is sometimes difficult to assert the true reality of TRD, as some factors, such as biased recruitments or effects on fertility, cannot always be ruled out. However, a few examples of human genetic diseases with repeated observations of an over-transmission of the mutant allele at the expense of the wild-type allele have been reported, several of them involving trinucleotides repetitions. For most trinucleotide repeat diseases, the larger the expansion is, the earlier the age at onset and the more severe the clinical symptoms become. In addition, the risk of expansion during meiosis generally increases with the number of repeats, resulting in the expansion of the length of trinucleotide repeats across generations, and thus the observation of more severe phenotypes and/or earlier ages of onset in successive generations, a process which is known as anticipation. Contractions of trinucleotide repeat numbers during meiosis have also been reported but are less frequent compared to expansions [130]. The instability of repeat expansions during meiosis and transmission of alleles with larger expansions to the offspring is thus influenced by multiple factors, including the size of the repeat, its structure and the sex of the transmitting parent. 

#### 4.3.1. Machado–Joseph Disease/Spinocerebellar Ataxia Type 3

Segregation studies have been undertaken concerning Machado–Joseph disease (also called spinocerebellar ataxia type 3 (MJD/SCA3)), the most common form of SCA. This autosomal dominant neurodegenerative disease is caused by an expansion of the number of CAG repetitions in exon 10 of the *MJD1* gene located on chromosome 14. The normal allele contains 11 to 44 CAG repeats, while the expanded allele contains at least 51 repeats. The expanded repeat is located in the open reading frame and produces an increase in the polyglutamine tract in ataxin-3 protein, causing the aggregation of the protein, which in turn is responsible for neuronal toxicity and subsequent neurodegeneration. Several studies have shown that the segregation is in favour of the expanded allele over the normal *MJD1* allele when the transmission is maternal [131]. The analysis of segregation in carrier sperm has been inconclusive thus far. It has been reported to follow non-Mendelian law in two studies on Japanese patients, with a higher degree of transmission of the mutant expanded allele in male meiosis [132,133], but this observation was not confirmed in another study, this time on French patients [134]. 

In parallel, three studies have evaluated the segregation of alleles with larger versus smaller CAG repeat numbers in the *MJD1* gene in normal heterozygous individuals [135,136,137], which revealed a tendency towards the preferential segregation of the shortest allele in maternal transmissions for the three studies. However, the results obtained for male meiosis were different: no bias was found in paternal transmission for one study [135], whereas similar results as for maternal transmissions were obtained in the two other studies [136,137]. Thus, there appears to be a selection of the two CAG alleles of opposite length not only depending on the sex, but also on the cellular context: either the shortest one in the absence of the disease-causing expanded allele or the longest one in heterozygous carrier fathers (for review, see [138]).

#### 4.3.2. Myotonic Dystrophy Type 1

Another example of trinucleotide repeat disease involving a TRD is myotonic dystrophy type 1 (DM1), an autosomal dominant dystrophy which mainly affects the neuronal and muscular systems, leading to abnormal cardiac conduction, respiratory defects and cataracts. In the most severe form of congenital myotonic dystrophy (CDM1), which is associated with a high number of repeats (>1000), intellectual disability may also occur. This condition results from a CTG expansion in the 3′ UTR of the *DMPK* gene (19q13.3). When the expansion extends beyond 300 repeats, it is responsible for abnormal CpG methylation, which can be gender-, age- or tissue-dependent. In the most severe form (CDM1), the defect is almost exclusively maternally transmitted. It was first hypothesized that it was because affected men often have oligozoospermia or azoospermia, which can lead to difficulties in conceiving, and the intellectual impairment often present makes it difficult to relate to a partner. However, these two factors cannot be held entirely responsible for the almost exclusively maternal transmission. Several other models have been proposed, such as a higher tendency to contractions in the male germline [139] or a selection against hypermethylated large expansions in the germline of male patients [140], which would be consistent with the oligozoospermia or azoospermia often reported in these patients.

#### 4.3.3. ARX

Another human mutation that has been reported to show a TRD is a polyalanine expansion in the *ARX* (*Aristaless-related homeobox gene*) transcription factor coding-gene located on the X chromosome [141]. This gene plays important roles in brain development and is associated with several neurodevelopmental diseases depending on the severity of the mutations [142]. The most frequent mutation of this gene is a duplication of a 24-base pair (c.429_452dup24) leading to a polyalanine expansion, which results in a partial loss-of-function phenotype characterized by intellectual disability and fine motor defects in males (Partington syndrome [MIM 309510]). Heterozygous females have no phenotype. Interestingly, clinicians have noted an excess number of affected males among the offspring of carrier females, raising the possibility of a TRD for this mutation ([141,143]). However, further studies were hampered by the relatively small number of affected families in the world and by their small size.

A mouse model was recently generated for this mutation, which recapitulates several phenotypic aspects observed in patients, such as memory and fine motor defects [144]. Interestingly, the analysis of the genotypes generated when breeding this mouse line reveals a TRD: female carriers transmit the mutated allele more often than expected following Mendelian’s law of segregation (χ^2^ = 7.2, *p* < 0.01 when female carriers are mated with wild-type males and χ^2^ = 10.97, *p* < 0.0009 when female carriers are mated with mutated males, Figure 3). On the opposite side, males carrying the mutation transmit the mutated allele as often as their Y chromosome, suggesting that the TRD probably occurs through female meiosis.

One hypothesis to explain this type of TRD is a similar mechanism of segregation that was reported by Wu et al. [37] concerning the preferential transmission of the *DDK* allele over the wild-type allele at the mouse *Om* locus (see “Ovum mutant” part). These authors demonstrated that this feature was due to a meiotic drive in the second meiotic division. Moreover, Wu et al. found that this TRD only occurred when the dyad in the oocyte II was heteromorphic. This means that, during the first meiosis division, a crossing-over occurred between the centromere and the *Om* locus only on one chromatid, but not on the *Om* locus of the other chromatid (Figure 4). Thus, during the second meiotic division, the chromatid carrying the *DDK* allele was preferentially transmitted to the ootid, while the chromatid carrying the wild-type allele went to the second polar body [2,37]. The mechanism underlying this asymmetry is still unknown. 

Alternatively, it is possible that the TRD observed for the ARX dup24 mutation is directly due to the partial loss-of-function effect of this mutation. Indeed, ARX is also strongly expressed in neuronal progenitors during development and *Arx* knock-down leads to a depletion of produced neurons in mice and humans [145,146], suggesting a role either in cell cycle progression and/or in the control of symmetric versus asymmetric divisions. As mice are a powerful model system to investigate the mechanisms and consequences of TRD [147], we have a unique model to investigate in depth the mechanisms of the TRD for *ARX* dup 24 mutation, which appears to be conserved between mouse and human, as shown in Figure 3.

## 5. Conclusions

Transmission ratio distortion appears to be a common phenomenon, and has been reported in a wide range of plants, fungi and animals [60]. In the latter, some of the most notable and scientifically challenging examples come from mice, livestock species and humans. In the house mouse, close to a century of research on TRD systems has largely contributed to our fundamental understanding of the genetic, molecular and evolutionary mechanisms involved. For instance, studies on the t-haplotype have helped to establish that TRD often results from complex interactions between multiple genetic factors, and that transmission distortion can be an important force shaping genetic architecture. More generally, the numerous known TRD systems in mice have led to a better understanding of the basic features of both female and male gametogenesis. Given the wide set of genetic and molecular tools available in this model species, the house mouse will certainly continue to be a valuable asset to dissect the mechanisms and consequences of transmission distortion. In contrast, knowledge about the prevalence and causes and consequences of TRD in livestock species and humans in still in its infancy. Nevertheless, in recent years, livestock species have emerged as great potential study systems for TRD. Large genomic and cytogenetic programs have allowed the characterization of several deviations from Mendelian transmission in cattle, pigs and horses. Additionally, they also provide the advantage of allowing researchers to perform largescale outcrossing experiments, ideal for exposing ancestrally active transmission distorters by decoupling drive elements and their suppressors. Finally, in humans, evidence of TRD remains fairly elusive, notably because of the difficulty of detecting drive in small families. Research in this area has been mainly driven by heath and fertility studies, given that TRD is known to induce reproductive defects in other species, and is already strongly suspected to be responsible for pathologies and fertility disorders in humans. Undoubtedly, future research in mice, livestock animals and other mammalian species will shed light on the prevalence and consequences of TRD in humans, thus allowing for better genetic counselling and lead to a better care for patients and their families.

## Figures and Tables

**Figure 1 ijms-24-01600-f001:**
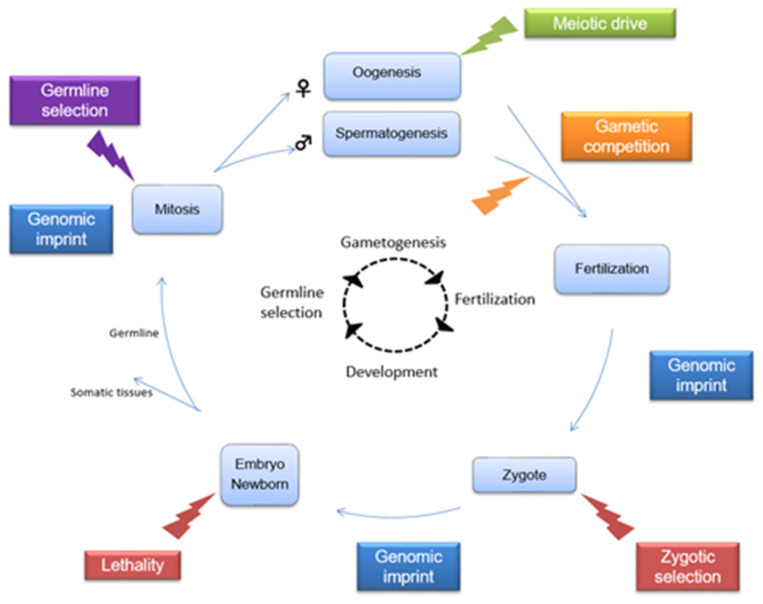
The different biological mechanisms inducing a transmission ratio distortion ranging from gamete formation to lethality at post-natal stages: Germline selection (purple box): During mitosis, mutations, mitotic gene conversion and crossing-overs can occur, causing a germline selection prior to meiosis. Meiotic drive (green box): Selfish genetic elements (meiotic drivers) can bias the normally random segregation of chromosomes in asymmetric division, such as in oogenesis. Gametic competition (yellow box): Before fertilization, there is a competition in mobility and/or viability and/or fertilization ability between spermatozoa. Genomic imprint (blue boxes): Errors in imprints in the parental germline or at fertilization or during the embryonic development can induce TRD. Zygotic selection or embryo/newborn lethality (red boxes): If the genotype is deleterious, it may lead to differential survival of the zygote, the embryo or even of the newborn.

**Figure 2 ijms-24-01600-f002:**
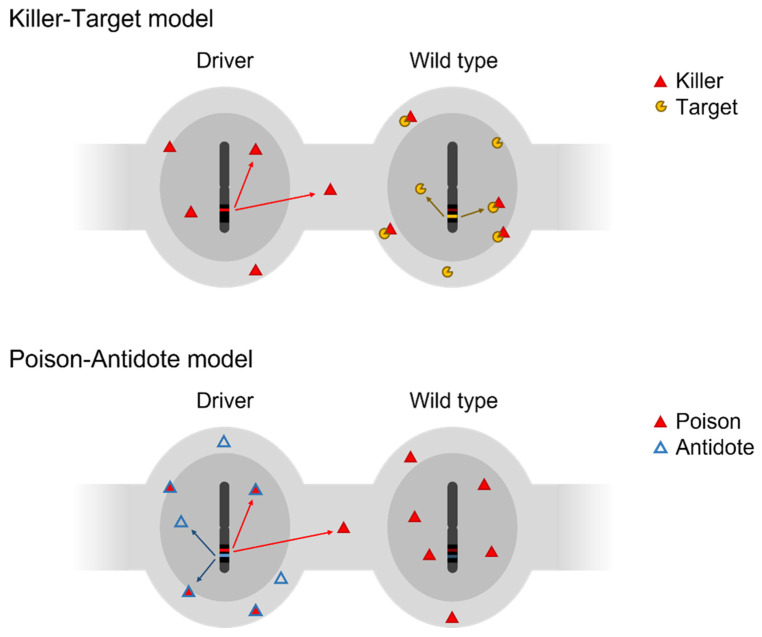
Killer–Target and Poison–Antidote models of transmission ratio distortion. The cells represented are round spermatids developing in a syncytium in a male heterozygous for the drive locus (black region on the chromosome). The killer/poison factors, produced solely by the « driver » chromosome, are shared across cytoplasmic bridges. The target and antidote factors are produced by the wild type and driver chromosomes, respectively, and are not shared across bridges.

**Figure 3 ijms-24-01600-f003:**
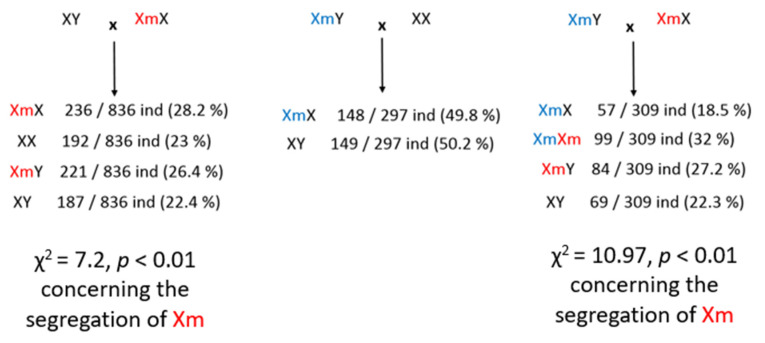
Details of the number of individuals obtained with different matings of the *Arx^dup24/0^* mice bearing a duplication of 24 pb in the *ARX* gene located on chromosome X. These matings reveal a TRD for the transmission of the maternal X mutated (red Xm) allele, but not of the paternal (blue) Xm allele.

**Figure 4 ijms-24-01600-f004:**
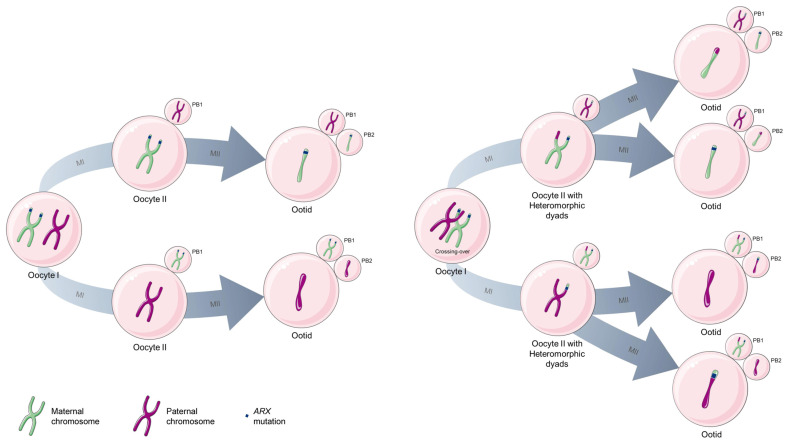
Oogenesis in heterozygous females for the *ARX dup24* mutation. Without crossing-over between the centromere and the *ARX* locus (**left**) and with a crossing-over between the centromere and the *ARX* locus (**right**). During the first meiotic division (M1), there is a reductional division leading to the separation of homologous chromosomes. There are two possibilities: the maternal chromosome segregates in oocyte II while the paternal one segregates to the first polar body (PB1), or vice versa. During the second meiotic division (M2), there is an equational division, with separation of sister chromatids. One of the chromatids segregates in the ootid while the other one segregates to the second polar body (PB2).

## Data Availability

All data are available on request.

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
