# Peer review of "Bypassing Mendel’s First Law: Transmission Ratio Distortion in Mammals"

_ijms, 2023, doi:10.3390/ijms24021600_

Round 1

Reviewer 1 Report

It is a very well-written review on a very interesting subject. TRD is a phenomenon of great importance for cell biology, genetics, evolutionary biology, reproductive medicine, and animal husbandry. The authors carefully collected and presented a large body of studies concerning the phenomenology and experimental analysis of TRD and a wide variety of its mechanisms.

The paper will certainly have a wide readership and, consequently, a high citation score. I am sure it deserves to be published in the International Journal of Molecular Sciences.

There are some minor comments that might help the authors improve the paper.

I would suggest creating a more attractive title than "Current knowledge on transmission bias in mouse, human and other Mammals". All review papers are aimed at presenting current knowledge about the subject of the review. Transmission bias is a rather vague expression. Transmission of what? It should be clear from the title. In general, it is an unnecessary and misleading synonym of genetic segregation distortion. I would remove mice and humans from the title. They are just better studied than other mammals, but they do not represent some special aspects of  TRD.

The subdivision of the review by these subjects is artificial, confusing, and leads to multiple repetitions. For example, Robetrsonian fusions are first mentioned on line 353 (TRD in mice), then on lines 467–468 (TRD in shrews and other mammals),  and again on lines 483–485 (TRD in humans), although the mechanisms of TRD are probably the same in all species.

The authors made an excellent introduction, briefly explaining when and how TRD can occur during development. I would suggest organizing the review according to this logic, classifying TRD cases by the developmental stage, at which they occur, rather than by  species. It would make the text easier to follow and remove unnecessary repetitions.

The abstract should be rewritten. Now it looks more like a table of contents than a summary or digest of the various mechanisms and consequences of TRD.

Line 40 "However, several situations in which this principle is not applied have been observed: we then talk about transmission bias, also called transmission ratio distortion (TRD)". This is too vague and colloquial for the definition of the main subject of the paper; please consider rewriting.

Lines 55–62: The letters a–e indicated in the caption for Fig. 1 are absent from the figure itself.

Line 60, "Ge-nomic imprint" - misprint

Line 80: "Post-meiotic TRD cannot happen in females because meiosis in oogenesis finishes after fertilization." This is not true, remember Medea (Maternal effect dominant embryonic arrest). Yes, it is described in insects, but here the authors consider general mechanisms, not only mammalian ones.

"There is no evidence of a driving element capable of biased transmission during both divisions," says line 104. This is not the case; HSR drives to oocyte during  MI (non-crossover HSR-carrying chromosome) and MII (crossover HSR-carrying chromatid).

Agulnik, S., Agulnik, A., & Ruvinsky, A. (1990). Meiotic drive in female mice heterozygous for the HSR inserts on chromosome 1. Genetical Research, 55(2), 97-100. doi:10.1017/S0016672300025325

Line 210: Wrong term: "bivalent pair. Bivalent is the pair.

The quality of Fig. 2 is rather poor. Its caption is redundant. It rephrases what is already said in the paragraph above.

Lines 219-224. I think HSR deserves more attention. It shows drive in both divisions. Its TRD depends not only on the maternal genetic background, but also on the presence or absence of HSR in the incoming sperm!  It also shows many other very interesting peculiarities. Ruvinsky, A. Meiotic drive in female mice: an essay. Mammalian Genome 6, 315–320 (1995). https://doi.org/10.1007/BF00364793; Pomiankowski, A. and Hurst, L.D., 1993. Siberian mice upset Mendel. Nature, 363 (6428), pp. 396-397; Agulnik, S., Sabantsev, I., & Ruvinsky, A. (1993). Effect of sperm genotype on chromatid segregation in female mice heterozygous for aberrant chromosome 1. Genetical Research, 61(2), 97-100. doi:10.1017/S0016672300031190

Line 229. "chromo-some" - misprint

Line 611: Be careful with "we." The lists of authors in this and the cited paper have only one author in common.

Line 619. I do not find any mention of the meaning of blue and red colors in Fig. 4.

Line 665. I was surprised to see acknowledgement of the people involved in breeding and genotyping the Arx-strain in the review paper. Does it mean that the paper contains unpublished data? If so, it should be explicitly stated.

Line 650. The conclusion is very brief and not very informative. This long paper, loaded with very interesting facts and hypotheses, needs a more systematic conclusion. It would be great if the authors outlined here the main gaps of knowledge in the reviewed field and suggested directions for future researches. 

P.S. Lines 501-516 are copied from ref 119. Please, rewrite

Author Response

It is a very well-written review on a very interesting subject. TRD is a phenomenon of great importance for cell biology, genetics, evolutionary biology, reproductive medicine, and animal husbandry. The authors carefully collected and presented a large body of studies concerning the phenomenology and experimental analysis of TRD and a wide variety of its mechanisms. The paper will certainly have a wide readership and, consequently, a high citation score. I am sure it deserves to be published in the International Journal of Molecular Sciences. There are some minor comments that might help the authors improve the paper.

We thank the reviewer for his/her kind comments. We did our best to answer to all queries and now submit an improved manuscript which take into account the reviewers’ remarks.

I would suggest creating a more attractive title than "Current knowledge on transmission bias in mouse, human and other Mammals". All review papers are aimed at presenting current knowledge about the subject of the review. Transmission bias is a rather vague expression. Transmission of what? It should be clear from the title. In general, it is an unnecessary and misleading synonym of genetic segregation distortion. I would remove mice and humans from the title. They are just better studied than other mammals, but they do not represent some special aspects of TRD.

We thank the reviewer for his/her suggestion and acknowledge that the title of our manuscript could be improved to be both more precise and more appealing. Following the reviewer's advice, we reworded the title to: « Bypassing Mendel's first law: transmission ratio distortion in mammals ».

The subdivision of the review by these subjects is artificial, confusing, and leads to multiple repetitions. For example, Robetrsonian fusions are first mentioned on line 353 (TRD in mice), then on lines 467–468 (TRD in shrews and other mammals), and again on lines 483–485 (TRD in humans), although the mechanisms of TRD are probably the same in all species. The authors made an excellent introduction, briefly explaining when and how TRD can occur during development. I would suggest organizing the review according to this logic, classifying TRD cases by the developmental stage, at which they occur, rather than by species. It would make the text easier to follow and remove unnecessary repetitions.

We thank the reviewer for underlying the quality of the introduction. We understand the reviewer's point about the subdivision of the manuscript. In fact, we evaluated in the initial draft the possibility of subdividing the review as suggested by the reviewer. After collective discussion between the authors, we however judged that an organization "by species" would be more appropriate for this review, because as you mentioned at the beginning of your review, the readers will be of various specialties and our current organization seems more friendly to most of them. Following the reviewer's comment, we have again collectively evaluated this possibility but our feeling remains unchanged, therefore we would like to keep our current subdivision.

The abstract should be rewritten. Now it looks more like a table of contents than a summary or digest of the various mechanisms and consequences of TRD.

Following the reviewer's comment, we have completely rewritten the abstract. It now states: « Mendel’s law of segregation states that the two alleles at a diploid locus should be transmitted equally to the progeny. A genetic segregation distortion, also referred to as transmission ratio distortion (TRD), is a statistically significant deviation from this rule. TRD has been observed in several mammal species and may be due to different biological mechanisms occurring at diverse time-points ranging from gamete formation to lethality at post-natal stages. In this review, we describe examples of TRD and their possible mechanisms in mammals based on current knowledge. We first focus on the differences between TRD in male and female gametogenesis in the house mouse, in which some of the most well studied TRD systems have been characterized. We then describe known TRD in other mammals, with a special focus on the farmed species and in the peculiar common shrew species. Finally, we discuss TRD in human diseases. So far, to our knowledge, it is the first time that such description is proposed. This review will definitely help better comprehend the processes involved in TRD. A better understanding of these molecular mechanisms will imply a better comprehension of their impact on fertility and on genome evolution. At term, this should allow a better genetic counseling and lead to a better care for families in humans. »

Line 40 "However, several situations in which this principle is not applied have been observed: we then talk about transmission bias, also called transmission ratio distortion (TRD)". This is too vague and colloquial for the definition of the main subject of the paper; please consider rewriting.

We agree with the reviewer's comment and have reformulated the sentence which now states: « Exceptions in which a statistical deviation from the 1:1 Mendelian inheritance ratio is observed are grouped under the term transmission ratio distortion (TRD) and occur when one of the two alleles of one of the parents is preferentially transmitted to the offspring. »

Lines 55–62: The letters a–e indicated in the caption for Fig. 1 are absent from the figure itself.

We thank the reviewer for pointing this out. The legend in Figure 1 now identifies correctly the different key elements by their color.

Line 60, "Ge-nomic imprint" – misprint

This is now corrected.

Line 80: "Post-meiotic TRD cannot happen in females because meiosis in oogenesis finishes after fertilization." This is not true, remember Medea (Maternal effect dominant embryonic arrest). Yes, it is described in insects, but here the authors consider general mechanisms, not only mammalian ones.

We agree with the reviewer's comment and have now modified our sentence. It now states: "Although no post-meiotic TRD has been reported in mammalian oocytes, it is to note that it has been observed in insects."

Subsequently, we added a new reference to support our statement: Lorenzen MD, Gnirke A, Margolis J, et al. The maternal-effect, selfish genetic element Medea is associated with a composite Tc1 transposon. Proc Natl Acad Sci U S A. 2008;105(29):10085-10089. doi:10.1073/pnas.0800444105.

The reference list was modified accordingly.

"There is no evidence of a driving element capable of biased transmission during both divisions," says line 104. This is not the case; HSR drives to oocyte during  MI (non-crossover HSR-carrying chromosome) and MII (crossover HSR-carrying chromatid).

Agulnik, S., Agulnik, A., & Ruvinsky, A. (1990). Meiotic drive in female mice heterozygous for the HSR inserts on chromosome 1. Genetical Research, 55(2), 97-100. doi:10.1017/S0016672300025325

We agree with the reviewer's comment. The sentence was altered. It now states : « There is no evidence of a driving element capable of biased transmission during both divisions, except for the Homogenously stained region (HSR) element which is discussed below. »

Line 210: Wrong term: "bivalent pair. Bivalent is the pair.

This is now corrected.

The quality of Fig. 2 is rather poor. Its caption is redundant. It rephrases what is already said in the paragraph above.

We removed figure 2 following comments from both reviewers agreeing on its low quality and redudance. The numbering of the following figures has been changed accordingly.

Lines 219-224. I think HSR deserves more attention. It shows drive in both divisions. Its TRD depends not only on the maternal genetic background, but also on the presence or absence of HSR in the incoming sperm!  It also shows many other very interesting peculiarities. Ruvinsky, A. Meiotic drive in female mice: an essay. Mammalian Genome 6, 315–320 (1995). https://doi.org/10.1007/BF00364793; Pomiankowski, A. and Hurst, L.D., 1993. Siberian mice upset Mendel. Nature, 363 (6428), pp. 396-397; Agulnik, S., Sabantsev, I., & Ruvinsky, A. (1993). Effect of sperm genotype on chromatid segregation in female mice heterozygous for aberrant chromosome 1. Genetical Research, 61(2), 97-100. doi:10.1017/S0016672300031190

We thank the reviewer for the constructive suggestion. We have now expanded the section regarding HSR. We added a new paragraph and a new reference: Ruvinsky A. Meiotic drive in female mice: an essay. Mamm Genome. 1995;6(5):315-320. doi:10.1007/BF00364793

Reference list was modified accordingly.

Line 229. "chromo-some" – misprint

This is now corrected.

Line 611: Be careful with "we." The lists of authors in this and the cited paper have only one author in common.

Following reviewer's comment, we removed the word « we » from the whole paragraph.

Line 619. I do not find any mention of the meaning of blue and red colors in Fig. 4.

Thank you for pointing out this omission. Legend of figure 4 (now figure 3) was modified accordingly.

Line 665. I was surprised to see acknowledgement of the people involved in breeding and genotyping the Arx-strain in the review paper. Does it mean that the paper contains unpublished data? If so, it should be explicitly stated.

Yes, the data showing the TRD in Arx mutant mice are unpublished data. We now made it clear lines 690-692: “we here have a unique model to investigate in depth the mechanisms of the TRD for ARX dup 24 mutation which appears to be conserved between mouse and human, as shown here (Figure 3)”.

Line 650. The conclusion is very brief and not very informative. This long paper, loaded with very interesting facts and hypotheses, needs a more systematic conclusion. It would be great if the authors outlined here the main gaps of knowledge in the reviewed field and suggested directions for future researches.

We agree with the reviewer's comment. We have expanded the conclusion which is now presented as a new section. It now states : « Transmission ratio distortion appears to be a common phenomenon, and has been reported in a wide range of plants, fungi and animals [60]. In latter, some of the most notable and scientifically challenging examples come from mice, livestock species and humans. In the house mouse, close to a century of research on TRD systems has largely contributed to our fundamental understanding of the genetic, molecular and evolutionary mechanisms involved. For instance, studies on the t-haplotype have helped establishing that TRD often results from complex interactions between multiple genetic factors, and that transmission distortion can be an important force shaping genetic architecture. More generally, the numerous known TRD systems in mice have led to a better understanding of the basic features of both female and male gametogenesis. Given the wide set of genetic and molecular tools available in this model species, the house mouse will certainly continue to be a valuable asset to dissect the mechanisms and consequences of transmission distortion. In contrast, knowledge about the prevalence and causes and consequences of TRD in livestock species and humans in still in its infancy. Nevertheless, in recent years, livestock species have emerged as great potential study systems for TRD. Large genomic and cytogenetic programs have allowed the characterization of several deviations from mendelian transmission in cattle, pigs and horses. Additionally, they also provide the advantage of allowing to perform largescale outcrossing experiments, ideal to expose ancestrally active transmission distorters by decoupling drive elements and their suppressors. Finally, in humans, evidence of TRD remains fairly elusive, notably because of the difficulty of detecting drive in small families. Research in this area has been mainly driven by heath and fertility studies, given TRD is known to induce reproductive defects in other species, and is already strongly suspected to be responsible for pathologies and fertility disorders in humans. Undoubtedly, future research in mice, livestock animals and other mammalian species will contribute to shed light on the prevalence and consequences of TRD in humans, thus allowing a better genetic counselling and lead to a better care for patients and their families. »

P.S. Lines 501-516 are copied from ref 119. Please, rewrite

This part of the text has been modified: « A transmission bias has also been reported in genetic diseases caused by point mutations. For example, pseudohypoparathyroidism type 1A (PHP1A) and pseudopseudohypoparathyroidism (PPHP) are two dominant genetic diseases caused by rare loss-of-function mutations in the GNAS (Guanine Nucleotide Binding Protein, Alpha Stimulating Activity) gene, which encodes the α-subunit of the stimulatory G protein (Gsα). PHP1A is due to mutations in the maternal allele and results in Albright’s hereditary osteodystrophy (AHO) with hormonal resistance, whereas PPHP, characterised by AHO features but without hormonal resistance, is due to mutations in the paternal allele. These two different phenotypes are explained by imprinting of the GNAS gene, and thus of the expression of Gsα, which is different depending on the sex of the transmitting parent. For both diseases, a TRD has been recently shown with an excess of transmission of mutated alleles from the mother to the offspring whereas in contrast, a Mendelian distribution was observed when the mutations were paternally inherited [121] suggesting a role of Gsα in oocyte biology or embryogenesis. Indeed, several studies have shown that AMPc levels and the AMPc pathway-protein kinase A are involved in the regulation of female meiosis in mammals, playing a role in both ovaries by controlling meiotic prophase 1 in oocytes, folliculogenesis and asymmetrical division [122–125]. Since GNAS is maternally expressed in the ovaries [126], this could affect the transmission of this allele to the offspring ».

We again thank the reviewer for his/her time and constructive comments.

Reviewer 2 Report

This is a comprehensively written review providing a good overview over TRD with many examples.

Figures 2 and 5 should be improved (enlarge the display of chromosomes to show the segregation more clearly).

Line 417-419: the advantage of t haplotype sperm is not based on higher velocity, but on more progressive movement compared to + sperm; see Ref 5, referenced on line 417).

A few minor spelling mistakes occur, e.g. TDR on line 627

Author Response

This is a comprehensively written review providing a good overview over TRD with many examples.

We thank the reviewer for his/her kind comments.

Figures 2 and 5 should be improved (enlarge the display of chromosomes to show the segregation more clearly).

Figure 2 was removed following both reviewer’s comment on its quality and reviewer 1 comment on its redundancy.

Figure 5 (now figure 4) has been fully reworked to provide better reading comfort.

Line 417-419: the advantage of t haplotype sperm is not based on higher velocity, but on more progressive movement compared to + sperm; see Ref 5, referenced on line 417).

We agree with the reviewer's comment. The use of the words "higher velocity" was a shortcut and has been changed to "a more progressive movement" which is more appropriate.

A few minor spelling mistakes occur, e.g. TDR on line 627

Thank you for correcting this typo that we missed. It is now corrected.

We hope to have answered all queries and thank the reviewer to allowed us to now submit an improved manuscript.
